# Vitamin D and COVID-19 Severity in Hospitalized Older Patients: Potential Benefit of Prehospital Vitamin D Supplementation

**DOI:** 10.3390/nu14081641

**Published:** 2022-04-14

**Authors:** François Parant, Justin Bouloy, Julie Haesebaert, Lamia Bendim’red, Karine Goldet, Philippe Vanhems, Laetitia Henaff, Thomas Gilbert, Charlotte Cuerq, Emilie Blond, Muriel Bost, Marc Bonnefoy

**Affiliations:** 1Biology Center South, Hôpital Lyon Sud, 69310 Pierre-Bénite, France; francois.parant@chu-lyon.fr (F.P.); charlotte.cuerq@chu-lyon.fr (C.C.); emilie.blond@chu-lyon.fr (E.B.); muriel.bost@chu-lyon.fr (M.B.); 2Department of Geriatric Medicine, Groupement Hospitalier Sud, CHU de Lyon, 69495 Pierre-Bénite, France; thomas.gilbert@chu-lyon.fr (T.G.); marc.bonnefoy@chu-lyon.fr (M.B.); 3Department of Clinical Research and Epidemiology, Public Health Unit, Groupement Hospitalier Est, 69002 Lyon, France; julie.haesebaert01@chu-lyon.fr; 4RESHAPE Research on Healthcare Performance Inserm U1290, Université Lyon 1, 69008 Lyon, France; 5Clinical Research Centre, Ageing, Brain, Fragility-Hôpital des Charpennes, 69100 Villeurbanne, France; ext-lamia.bendimred@chu-lyon.fr (L.B.); karine.goldet@chu-lyon.fr (K.G.); 6Department of Hygiene, Epidemiology and Prevention, Hôpital Édouard Herriot, Hospices Civils de Lyon, 69003 Lyon, France; philippe.vanhems@chu-lyon.fr (P.V.); laetitia.henaff@chu-lyon.fr (L.H.); 7ICIR-International Center for Infectiology Research (Team PHE3ID), Claude Bernard Lyon 1 University, Inserm, U1111, CNRS, UMR5308, ENS Lyon, 46 Allée d’Italie, 69007 Lyon, France

**Keywords:** COVID-19, elderly, vitamin D, supplementation, mortality

## Abstract

Studies involving the associations between vitamin D supplementation taken before the onset of COVID-19 infection and the clinical outcomes are still scarce and this issue remains controversial. This study aimed to assess the relationships between vitamin D (VitD) status and supplementation and coronavirus disease 2019 (COVID-19) severity in older adults (average age of 78 years) hospitalized for COVID-19. We conducted an observational retrospective cohort study with 228 older hospitalized patients during the first wave of the COVID-19 pandemic. The outcomes were in-hospital mortality secondary to COVID-19 or critically severe COVID-19. A logistic regression analysis was conducted to test whether pre-hospital VitD supplementation was independently associated with severity. In this study, 46% of patients developed a severe form and the overall in-hospital mortality was 15%. Sixty-six (29%) patients received a VitD supplement during the 3 months preceding the infection onset. Additionally, a VitD supplement was associated with fewer severe COVID-19 forms (OR = 0.426, *p* = 0.0135) and intensive care unit (ICU) admissions (OR = 0.341, *p* = 0.0076). As expected, age > 70 years, male gender and BMI ≥ 35 kg/m^2^ were independent risk factors for severe forms of COVID-19. No relationship between serum 25(OH)D levels and the severity of the COVID-19 was identified. VitD supplementation taken during the 3 months preceding the infection onset may have a protective effect on the development of severe COVID-19 forms in older adults. Randomized controlled trials and large-scale cohort studies are necessary to strengthen this observation.

## 1. Introduction

Advanced age is by far the greatest risk factor for COVID-19-related death, irrespective of underlying co-morbidities [1]. It is not known why older adults are more vulnerable to COVID-19, but hypotheses are emerging, such as immune senescence and chronic low-grade inflammation, termed “inflamm-aging” [2,3,4]. A key issue is to understand what the roles of nutritional status and levels of nutrients are in supporting the immune functions of older patients with COVID-19 [5,6]. Multiple arguments suggested the involvement of vitamin D in reducing the risk of SARS-CoV-2 infection and related severity [7,8,9]. It is well recognised that vitamin D plays numerous roles besides calcium–phosphorus metabolism, notably in modulating the adaptive and innate immune systems [10,11,12]. Regarding the fact that vitamin D deficiency is a general risk factor for acute respiratory infections (ARIs) and that supplementation may prevent ARIs [13,14], several authors advocated taking vitamin D supplements to reduce the effects of COVID-19 based on a risk/reward analysis [9,15]. However, according to the “NIH, Coronavirus Disease 2019 (COVID-19) Treatment Guidelines”, data are still insufficient to recommend either for or against the use of vitamin D supplementation for the treatment of COVID-19 [16]. Although “vitamin D and COVID-19” research is very active [17], whether on its status or its supplementation, current data are contradictory [18]. Based on a review of the literature, poor vitamin D status seems to be associated with an increased risk of contracting SARS-CoV-2 infection, whereas age, gender and comorbidities seem to play a more important role in COVID-19 severity and mortality [19]. Serum 25(OH)D is the major circulating form of vitamin D and a standard indicator of vitamin D status. Vitamin D deficiency has been variably defined. However, all guidelines unanimously agree that serum levels of 25(OH)D < 25 nmol/L (10 µg/L) should be avoided at all ages [20,21]. In older adults, risk factors contributing to vitamin D deficiency include reduced nutritional intake of vitamin D, obesity, limited time spent outdoors, and decreased cutaneous synthesis of vitamin D [22]. Vitamin D deficiency is common throughout the European population at prevalence rates that are concerning [23]. A French study conducted in people over 65 y/o in primary care has concluded that prescription of vitamin D supplements is still too scarce and should be encouraged [24]. Interestingly, a study by Annweiler et al. suggested that vitamin D supplementation taken regularly over the year preceding the diagnosis of COVID-19 might be associated with better survival in frail older patients hospitalized for COVID-19 [25].

However, these results, although very interesting and promising, still need to be confirmed. For this reason, we conducted a retrospective cohort study during the first wave of the pandemic to document the possible relationship between vitamin D, status and supplementation, and COVID-19 severity in hospitalised patients over 50 years old more fully. The objectives of our study were to identify a possible association between pre-hospital vitamin D supplementation and the development of severe forms of COVID-19, and to assess the frequency of vitamin D deficiency and its association with the severity of COVID-19 among elderly patients hospitalised for COVID-19.

## 2. Materials and Methods

### 2.1. Study Design and Participants’ Profile

The present study was a sub-study of the observational study “MicroCovAging” (micronutrients status in aging patients with COVID-19) (ClinicalTrials.gov Identifier: NCT04877509) conducted at the Hospices Civils de Lyon (HCL), in Lyon, France, and focused on the potential link between vitamin D and COVID-19 severity. Two hundred and thirty patients hospitalized in the Lyon public hospitals were consecutively included during the first wave of the pandemic (admission date from 1 March 2020 to 30 June 2020). Two patients have retroactively withdrawn their consent and were consequently excluded from the data analysis. All enrolled patients were aged ≥50 y/o and diagnosed with active SARS-CoV-2 infection by RT-PCR. The flow chart of the study is shown in Figure 1.

### 2.2. Data Collection

Patient demographics, clinical features and treatment information, including vitamin D supplementation, were prospectively collected by reviewing patient medical records and recorded in an anonymous inpatient COVID-19 database (NOSO-COR project; Hospices Civils de Lyon, Lyon, France) [26]. Levels of biochemical and haematological biomarkers of COVID-19 disease severity (CRP, LDH, albumin, leukocyte, neutrophilic polynuclear, lymphocyte, platelets, fibrinogen, D-dimer) were obtained by routine testing during hospitalisation of the patients.

### 2.3. Pre-Hospital Vitamin D Supplementation

Patients were classified in two groups depending on whether or not vitamin D supplementation was taken during the three months preceding the diagnosis of SARS-CoV-2 infection. The indications for supplementation of these patients followed the guidelines for musculoskeletal health, irrespective of the pandemic context. Pre-hospital vitamin D supplementation was evaluated as a potential dichotomous protective factor for severe COVID-19.

### 2.4. Serum Concentrations of 25(OH)D and Vitamin D Status

The status of vitamin D was evaluated by serum 25(OH)D concentrations measurement and classified as per French ORIG guidelines (deficient: <25 nmol/L (<10 µg/L), insufficient: 25–75 nmol/L (10–30 µg/L) and sufficient: ≥75 nmol/L (≥30 µg/L)) [21,27]. Two different time points were selected: (i) up to three months before the SARS-CoV-2 infection onset (T1), and (ii) during the first week of hospitalisation (T2). Serum concentrations of 25(OH)D obtained at T1 were considered as baseline values, whereas those obtained at T2 could be impacted by the COVID-19-associated hyper-inflammation [28]. The levels of 25(OH)D obtained before SARS-CoV-2 infection (T1), when available, were gathered via routine testing. The levels obtained during hospitalisation (T2) were both routine testing, and/or additional analyses were carried out as part of the MicroCovAging study, thanks to the COVID-19 biobank hosted at Biological Resource Centre (Hospices Civils de Lyon). This biobank consists of frozen tube bottoms coming from routine sampling during COVID-19 patient hospitalisations. All serum 25(OH)D analyses were performed by the multi-site medical biology laboratory “LBMMS” (University Hospital of Lyon) with a validated chemiluminescence immunoassay (LIAISON^®^ 25 OH Vitamin D TOTAL Assay). This assay detects 25-Hydroxy D2 as well as 25-Hydroxy D3. The relationship between serum 25(OH)D concentrations and clinical/biological COVID-19 severity was investigated.

### 2.5. Study Clinical Endpoints of COVID-19 Disease Severity

The clinical endpoints of the study were: (i) the in-hospital mortality directly related to COVID-19, (ii) the onset of severe forms of COVID-19, and (iii) the admission to an intensive care unit (ICU). The assignment of causes of death was carried out via a medical evaluation by two independent physicians. The severe forms of COVID-19 were defined as any of the following criteria: high oxygen requirement (i.e., oxygen flow > 5 L/min), intubation, or death related to COVID-19 during the hospital stay. ICU admission was not included in the criteria for severe forms of COVID-19 because all patients were ineligible for ICU admission.

### 2.6. Statistical Analysis

Since the quantitative variables distribution significantly deviated from normal distribution (*p* < 0.001) (except for BMI, fibrinogen and albumin), we expressed results as medians and interquartile ranges (Q1–Q3) and used a non-parametric test for all comparisons. Continuous data were compared using a Mann–Whitney *U* test, and categorical data were compared using a Fisher’s exact test. The French High Council for Public Health, in accordance with the literature and international databases, recommends that age ≥ 70 y/o and a BMI ≥ 35 kg/m^2^ or <18.5 kg/m^2^ should be considered at risk of severe COVID-19 [29]. That is why results were stratified according to the age of patients at diagnosis (<70 y/o or ≥70 y/o) and body mass index (BMI) of 18.5 kg/m^2^ (18.5–35 kg/m^2^) and ≥ 35 kg/m^2^. Correlations between variables were analysed using the Pearson correlation test. Logistic regression models were also performed to separately explain each clinical endpoints (death, severe forms or admission to an ICU) according to age (<70 y/o/≥70 y/o), gender (male/female), BMI (<35 kg/m^2^/≥35 kg/m^2^), and pre-hospital vitamin D supplementation (yes/no). All tests were two-tailed and the significance level was defined as a *p* value < 0.05. Statistical analyses were performed using SAS 9.4 software (2013, SAS Institute Inc., Cary, NC, USA).

### 2.7. Ethical Consideration

The study was submitted to the scientific and ethical committee of the Hospices Civiles de Lyon and was reported on the clinicaltrials.gov website under the number NCT04877509. Patients were informed of the study by their medical doctor during hospitalisation or by post, thanks to a generic information sheet dedicated to COVID-19 research at the Hospices Civils de Lyon. In the regulatory process, patients did not have to sign a consent form but did not have to object to the research within one month of receiving the information note. The processing of personal data carried out for this study falls within the scope of “Reference Methodology No. 4” (MR-004) of the Commission Nationale de l’Informatique et des Libertés (CNIL).

## 3. Results

### 3.1. Patient Demographics and Comorbidities

The baseline characteristics of the 228 enrolled patients are shown in Table 1. According to the inclusion criteria, all patients included were hospitalized, with SARS-CoV-2 infection confirmed by RT-PCR, and over 50 years old. The median age was 78 years (IQR, 68–87), with 43% of patients over 80 years old. One-quarter of the patients lived in institutions before hospitalisation. Regarding the risk factors for COVID-19 other than age, the sex ratio (M/F) was 1.3 (129/99) and 7% of the enrolled patients had a body mass index (BMI) greater than 35 kg/m^2^. The most common comorbidities were arterial hypertension (56%) and other cardiovascular diseases (36%). Multiple comorbidities were present in 63% of the patients.

### 3.2. Clinical Outcomes

One hundred and five patients (46%) developed a severe form of COVID-19 and the overall in-hospital mortality related to COVID-19 was 15% (Table 1). As expected, our data showed that age was associated with COVID-19 mortality (91% of deceased patients were over 70 years old, *p* = 0.004). Among patients who died, 28 were males and 7 were females (sex ratio = 4.0, *p* = 0.003). Male gender was also associated with both a higher risk for severe COVID-19 outcomes (sex ratio = 2.2, *p* < 0.001) and a higher risk for admission to ICU (sex ratio = 2.5, *p* < 0.001). Additionally, the proportion of obese patients with BMI ≥ 35 kg/m^2^ was significantly higher in patients with severe COVID-19 outcomes (12% vs. 3%, *p* = 0.02). Eighty-four patients (37%) were admitted to an ICU, and the ICU mortality related to COVID-19 was 23%. Sixteen deceased patients (16/35) died before ICU admission or were ineligible for ICU admission.

### 3.3. Laboratory Parameters

The maximum increases during hospitalisation of C-reactive protein (CRP), lactate dehydrogenase (LDH), leukocyte and neutrophil counts, platelets count, fibrinogen, and D-dimer were significantly higher in patients with severe COVID-19 compared to moderately ill patients, whereas the nadirs of lymphocyte count and albumin concentration were significantly lower (Table 2).

### 3.4. Pre-Hospital Vitamin D Supplementation

Sixty-six (29%) patients were identified as having received a vitamin D supplement during the three months preceding the diagnosis of SARS-CoV-2 infection with almost exclusively (64/66) cholecalciferol (vitamin D3). Large modalities of vitamin D supplementation were observed. Half of them had taken a vitamin D supplement every 2 or 3 months at a dosage of 80,000 or 100,000 IU and only 13% daily at a dosage of less than 1000 IU. As expected, patients who had received pre-hospital vitamin D supplementation were older than patients who were not supplemented (89 years vs. 74 years), *p* < 0.001. Female patients were more frequently supplemented than male patients (39% vs. 21%, *p* = 0.003). In addition, patients living in institutions were also more frequently supplemented (50%) than those living at home (22%) (*p* < 0.001).

### 3.5. Association between Pre-Hospital Vitamin D Supplementation and Clinical Outcomes

Table 3 shows the impact of pre-hospital vitamin D supplementation on COVID-19 severity and mortality. Overall, patients without vitamin D supplementation developed severe forms of COVID-19 (53% vs. 28%, *p* = 0.001) and were admitted to an ICU more frequently (46% vs. 15%, *p* < 0.001). Once stratified by gender and age (<70 y/o or ≥70 y/o), pre-hospital vitamin D supplementation showed no protective effect on disease severity except for admission to an ICU in the subgroup of female patients over 70 years old (*p* = 0.046).

The results of the logistic regression analysis (Table 4) suggest that BMI > 35 kg/m^2^, male gender and being >70 years old were significantly associated with an increased risk of mortality. Additionally, pre-hospital vitamin D supplementation was associated with a significant reduction in the occurrence of severe disease and ICU admission, but without significant effect on mortality.

### 3.6. Association between Serum Concentrations of 25(OH)D and Clinical Outcomes

A total of 106 serum 25(OH)D concentrations were obtained from 96 patients. Only thirty-seven serum 25(OH)D concentrations were available for the baseline time point T1 (median delay of sampling before admission: 35 days with an IQR of 51–12 days). Sixty-nine serum 25(OH)D concentrations were available for the time point T2 (i.e., during the first week of hospitalisation) with a median delay of sampling after admission of 2 days. Median serum 25(OH)D concentrations at time points T1 and T2 were similar between those who died and those who did not, between those who developed a severe form of COVID-19 and those who did not, and between patients admitted to an ICU and those who were not (Table 5). Due to the small sample size, the stratification by age and gender could not be performed.

### 3.7. Association between Vitamin D Status and Clinical Outcomes

A vitamin D deficiency was found in 10% of the patients, and a vitamin D insufficiency was found in 60% of the patients. Concerning the relationship between vitamin D status and COVID-19 outcomes, we were unable to demonstrate any relationship between 25(OH)D deficiency or insufficiency and worse COVID-19 outcomes (Table 6).

## 4. Discussion

This observational study involving 288 patients, with an average age of 78 years old, hospitalized in Hospices Civils de Lyon hospitals aimed to investigate the possible link between vitamin D status and supplementation, and COVID-19 severity in a population representative of French aging patients hospitalized for COVID-19. The observed sex ratio of 1.3 was expected and consistent with patients hospitalized for COVID-19 in France [30]. The well-known risk factors for severe COVID-19, such as male gender, age and overweight were also found in our study. Likewise, the levels of the inflammatory biomarkers were related to the severity of the disease. The number of patients admitted to an ICU (37%) was slightly higher than that described in the literature (generally reported around 20%) [31,32], probably due to the older age and to the multiple comorbidities. However, only 20% of the over 70 year olds and only 5% of the over 80 years olds hospitalized for COVID-19 were admitted to an ICU in line with all French patients hospitalized for COVID-19 in the first wave [32]. The in-hospital mortality was 15% and 53% of those over 80 years old, in agreement with data published elsewhere [30,32].

Since the COVID-19 pandemic started, some investigations have been devoted to the study of a potential beneficial effect of vitamin D on the disease course. COVID-19 infection leads to a systemic inflammatory reaction with notably increased C-reactive protein (CRP), leukocytes and fibrinogen, associated with lymphocyte consumption, a reaction that is emphasized in case of severe illness [33]. Vitamin D has been described as a factor able to lower inflammation [10,11,12]. Vitamin D has diverse and widespread effects on the immune system, due to the expression of its receptor in most immune cells [34] and plays a role in the maintenance of immune homeostasis. However, the mechanisms behind are unclear. Conflicting results on the effects of vitamin D on immune cells are well described in the meta-analysis by Greiller and Martineau [11]. In a recent review focused on lung-inflammation related to COVID-19, Fakhoury et al. concluded that the anti-inflammatory and anti-thrombotic effects of vitamin D are promising features, suggesting efficiency against immunothrombosis related to the disease [35]. However, the clinical benefit of vitamin D therapy as an immunomodulator has still not been demonstrated. By investigating different factors influencing cytokine production in a large cohort of healthy individuals, ter Horst et al. found that variations in circulating vitamin D concentrations have only a limited effect on the different cytokines production systems [36]. These results suggest that vitamin D is probably one factor among others that contributes to modulating the immune system and to reducing the inflammatory response.

These data do not rule out the fact that low 25(OH) vitamin D concentrations are generally associated with a higher risk of SARS-CoV-2 infection [37]. However, less is known about low 25(OH) vitamin D concentrations and severe forms of COVID-19. A positive association between vitamin D deficiency (<20 ng/mL) and severity of the disease and mortality has been identified in a meta-analysis conducted in 2020 by Pereira et al. [38]. In 2021, Barassi et al. noticed that vitamin D deficiency was more frequent in patients on a continuous positive airway pressure device (CPAP) or on non-invasive mechanical ventilation (NIMV) [39]. However, in a recent meta-analysis on a large number of COVID-19 patients, Bassatne et al. concluded that no clear association could be identified between vitamin D deficiency (<20 ng/mL) and COVID-19-related health outcomes, such as mortality, ICU admission and invasive or non-invasive ventilation [40]. In our study, no significant association between low serum 25(OH) vitamin D concentrations and higher risk of developing severe forms of COVID-19 could be found, but our results lack statistical power.

Another question raised is that of a potential therapeutic or preventive action of vitamin D on COVID-19 severity. Assuming vitamin D supplementation to be effective as an adjunct treatment for SARS-CoV-2 infection, the pharmacokinetics of the vitamin D supplementation is unfavourable as acute phase treatment. After high-dose vitamin D supplementation, Fassio et al. [41] showed that maximum serum 25(OH)D concentrations were reached beyond 60 days after the beginning of the supplementation. Therefore, vitamin D supplementation started during COVID-19 infection has probably no effect on the course of the acute disease, whatever the dose. Other authors (Murai [42], Annweiler et al. [25], Leaf [43]) were also unable to find a positive effect when supplementation was started at the time of SARS-CoV-2 infection. On the contrary, in terms of preventing action, Annweiler et al. [25] found a significant decrease in 14-day mortality in a group of hospitalized frail elderly COVID-19 patients that had been regularly vitamin D supplemented for at least one year before infection, suggesting a beneficial effect when vitamin D is supplied for a longer period of time. Our results corroborate this. Despite a small number of cases, patients supplemented with vitamin D during the 3 months preceding the diagnosis of SARS-CoV-2 infection resulted in a significantly reduced occurrence of severe disease and ICU admission.

### Limitations and Strengths of the Study

Several limitations of our study should be acknowledged. First, the retrospective and observational nature of the study, as well as the limited number of patients, do not allow for any formal conclusion to be reached. Second, modalities of vitamin D supplementation (dosage and frequency of administration) were heterogeneous from one subject to another, despite guidelines. This is attributable to the observational nature of the study with real-life data, which is attested by the different prescriptions. This could reduce the generalisability of the results, but does not bring into question the beneficial effect of supplementation when provided on a long-term basis.

Nevertheless, several points make this study original. We took a snapshot of the vitamin D status of patients infected with SARS-CoV-2 and analysed the course of the disease depending on whether vitamin D supplementation or not was taken during the three months preceding the infection diagnosis. Moreover, the analysis of an elderly or even very elderly population, which is rarely represented in the literature, also makes our work interesting. To our knowledge, only one study has looked at the impact of pre-hospitalization supplementation in the very elderly [27]. As previously mentioned, the population of our study is representative of patients hospitalised for COVID-19 and of patients presenting severe forms of the disease during the first wave of the COVID-19 pandemic. Additionally, the fairly comprehensive nature of the analyses carried out have provided additional information to support the existing data.

## 5. Conclusions

Interestingly, vitamin D supplementation taken before the onset of the disease seems to have a protective effect on the development of severe forms of COVID-19 in patients aged over 70 years. However, our study reports no relationship between serum 25(OH)D levels and the severity of SARS-CoV-2 infection, which needs to be confirmed in a larger cohort. These new findings are further reasons to make vitamin D supplementation widespread, especially in this population at high risk of deficiency and under-supplementation. Randomised controlled trials and large-scale cohort studies are needed to reinforce this observation. Vitamin D supplementation needs to be more systematic while limiting expectations of its own effects.

## Figures and Tables

**Figure 1 nutrients-14-01641-f001:**
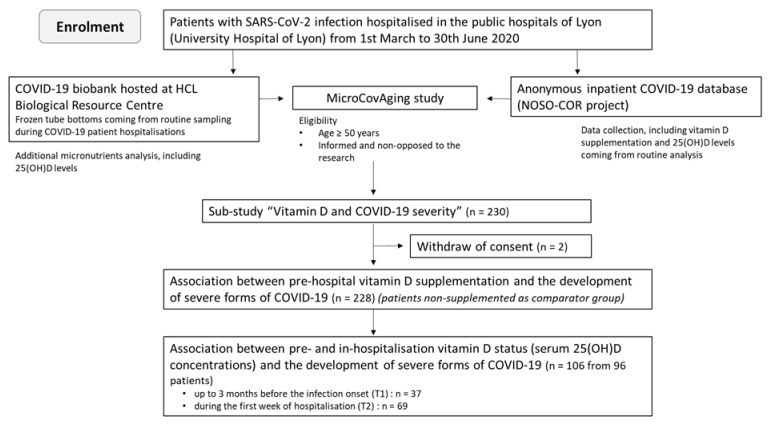
Flow chart summarizing the enrolment of patients included in the study.

**Table 1 nutrients-14-01641-t001:** Baseline demographics, comorbidities and clinical outcomes.

	Total Patients	COVID-19 Survivors	COVID-19-Related Death	Severe COVID-19 (Including Death) *	ICU Admission **
	(n = 228)	85% (193/228)	15% (35/228)	46% (105/228)	37% (84/228)
Median age (IQR)	78 years (68–87)	76 years (66–87)	82 years (76–86)	73 years (66–82)	71 years (64–77)
Age ≥ 70 years % (n)	72% (164/228)	68% (132/193)	91% (32/35)	67% (70/105)	56% (47/84)
Age ≥ 80 years % (n)	43% (99/228)	41% (80/193)	54% (19/35)	27% (28/105)	14% (12/84)
Sex ratio (male/female)	1.3 (129/99)	1.1 (101/92)	4.0 (28/7)	2.2 (72/33)	2.5 (60/24)
Living in institutions % (n)	25% (56/228)	24% (47/193)	26% (9/35)	19% (20/105)	6% (5/84)
Median BMI (IQR)	25.7 kg/m^2^ (22.1–29.1) n = 224 ***	25.9 kg/m^2^ (22.2–29.1) n = 190	24.7 kg/m^2^ (20.6–27.6) n = 34	26.7 kg/m^2^ (23.8–30.4) n = 104	27.3 kg/m^2^ (24.5–31.2) n = 84
BMI < 18.5 kg/m^2^ % (n)	8% (19/224)	8% (15/190)	12% (4/34)	5% (5/104)	2% (2/84)
BMI ≥ 35 kg/m^2^ % (n)	7% (16/224)	6% (11/190)	15% (5/34)	12% (12/104)	12% (10/84)
Arterial hypertension % (n)	56% (127/228)	54% (104/193)	66% (23/35)	59% (62/105)	56% (47/84)
Cardiac diseases % (n)	36% (83/228)	35% (68/193)	43% (15/35)	29% (30/105)	25% (21/84)
Diabetes mellitus (types 1 & 2) % (n)	32% (74/228)	34% (65/193)	26% (9/35)	33% (35/105)	36% (30/84)
Neuro-muscular diseases % (n)	19% (43/228)	21% (40/193)	9% (3/35)	12% (13/105)	13% (11/84)
Neuro-cognitive disorders % (n)	19% (43/228)	19% (37/193)	18% (6/35)	10% (11/105)	1% (1/84)
Renal diseases % (n)	14% (31/228)	14% (27/193)	11% (4/35)	9% (9/105)	8% (7/84)
Pulmonary diseases % (n)	13% (30/228)	12% (24/193)	17% (6/35)	15% (16/105)	11% (9/84)
Solid-organ cancers (diagnosed < 1 year ago) % (n)	8% (19/228)	8% (15/193)	11% (4/35)	8% (8/105)	5% (4/84)
Immunodeficiency diseases % (n)	7% (15/228)	6% (12/193)	9% (3/35)	9% (9/105)	8% (7/84)
Haematological cancers (diagnosed < 1 year ago) % (n)	4% (9/228)	4% (8/193)	3% (1/35)	2% (2/105)	1% (1/84)
Multiples comorbidities (≥2) % (n)	63% (144/228)	63% (122/193)	63% (22/35)	57% (60/105)	52% (44/84)

Abbreviations: ICU, intensive care unit; IQR, interquartile range; BMI, body mass index. * One of the following criteria: high oxygen requirement, intubation, or death related to COVID-19 during the hospital stay. ** All patients with severe forms of COVID-19 were ineligible for ICU admission. *** BMI was not available in 4 patients.

**Table 2 nutrients-14-01641-t002:** Biomarkers related to COVID-19 severity.

	Severe COVID-19, Including Death (n = 105)	Non-Severe COVID-19 (n = 123)	*p* Value **
CRP (mg/L) *	179 (111–271) (n = 104)	81 (46–140) (n = 122)	<0.001
LDH (U/L) *	352 (269–488) (n = 23)	240 (198–281) (n = 27)	<0.001
Leukocyte (giga/L) *	15.5 (11.4–19.1) (n = 105)	9.25 (7.10–11.61) (n = 121)	<0.001
Neutrophilic polynuclear (giga/L) *	12.6 (9.21–15.5) (n = 105)	7,09 (4.78–9.24) (n = 121)	<0.001
Lymphocyte (giga/L) *	0.5 (0.41–0.77) (n = 105)	0.77 (0.54–1.16) (n = 121)	<0.001
Platelets (giga/L) *	422 (330–541) (n = 105)	353 (284–450) (n = 121)	0.003
Fibrinogen (g/L) *	7.61 (6.32–9.17) (n = 103)	5.41 (4.08–7.14) (n = 104)	<0.001
D-dimer (μg/L) *	1879 (1136–5123) (n = 93)	1170 (673–2407) (n = 58)	0.004
Albumin (g/L) *	19.7 (16–23.9) (n = 98)	28.5 (25.4–32.8) (n = 119)	<0.001

Abbreviations: IQR, interquartile range; CRP, C-reactive protein; LDH, Lactate dehydrogenase. * Maximum values during hospitalisation for CRP, LDH, D-dimer, leukocyte, neutrophilic polynuclear, and platelets counts. Minimum values for lymphocyte count and albumin. ** Mann–Whitney *U* test.

**Table 3 nutrients-14-01641-t003:** Impact of the pre-hospital vitamin D supplementation on COVID-19 severity according to the study subgroups.

Subgroups	Pre-Hospital Vitamin D Supplementation	COVID-19-Related Death (n)			Severe COVID-19, Including Death (n)			ICU Admission (n)		
		No	Yes	*p* value *	No	Yes	*p* value *	No	Yes	*p* value *
All patients	No	134	28	0.231	76	86	0.001	88	74	<0.001
	Yes	59	7		47	19		56	10	
	Total (n)	193	35		123	105		144	84	
Stratified by gender										
Males	No	80	22	1.000	42	60	0.198	50	52	0.054
	Yes	21	6		15	12		19	8	
Females	No	54	6	0.240	34	26	0.010	38	22	<0.001
	Yes	38	1		32	7		37	2	
Stratified by gender and age										
Males ≥ 70 years	No	44	21	0.608	27	38	0.245	36	29	0.229
	Yes	19	6		14	11		18	7	
Males < 70 years	No	36	1	1.000	15	22	1.000	14	23	1.000
	Yes	2	0		1	1		1	1	
Females ≥ 70 years	No	33	4	0.358	23	14	0.121	28	9	0.046
	Yes	36	1		30	7		35	2	
Females < 70 years	No	21	2	1.000	11	12	0.480	10	13	0.220
	Yes	2	0		2	0		2	0	

* Fisher’s exact test.

**Table 4 nutrients-14-01641-t004:** Logistic regression analysis of association between clinical outcomes and covariates: impact of the pre-hospital vitamin D supplementation on COVID-19 severity.

**Death (n = 35/228) (Yes/No)**								
	univariate				multivariate			
Variable	Crude OR	95% CI		*p*-value	Crude OR	95% CI		*p*-value
BMI *	2.806	0.909	8.665	0.0729	4.727	1.248	17.911	0.0223
Age **	4.929	1.453	16.725	0.0105	7.586	2.083	27.627	0.0021
Sex ***	0.275	0.114	0.659	0.0038	0.263	0.102	0.672	0.0053
VD supp	0.568	0.235	1.373	0.2090	0.448	0.167	1.207	0.1124
**Admission to an ICU (n = 84/228) (Yes/No)**								
	univariate				multivariate			
Variable	Crude OR	95% CI		*p*-value	Crude OR	95% CI		*p*-value
BMI *	3.017	1.055	8.632	0.0395	2.948	0.923	9.418	0.0681
Age **	0.293	0.161	0.535	<0.0001	0.386	0.200	0.745	0.0046
Sex ***	0.368	0.207	0.654	0.0007	0.373	0.198	0.702	0.0022
VD supp	0.212	0.101	0.445	<0.0001	0.341	0.155	0.751	0.0076
**Severe form (n = 105/228) (Yes/No)**								
	univariate				multivariate			
Variable	Crude OR	95% CI		*p*-value	Crude OR	95% CI		*p*-value
BMI *	3.783	1.181	12.117	0.0251	4.017	1.179	13.689	0.0262
Age **	0.617	0.345	1.104	0.1035	0.841	0.442	1.599	0.5972
Sex ***	0.396	0.230	0.682	0.0008	0.414	0.232	0.739	0.0029
VD supp	0.357	0.193	0.661	0.0011	0.426	0.217	0.838	0.0135

Abbreviations: ICU, intensive care unit; OR, Odds ratio; BMI, body mass index; VD supp, pre-hospital vitamin D supplementation. * BMI > 35 kg/m^2^; ** Age > 70 years old; *** Male.

**Table 5 nutrients-14-01641-t005:** Serum 25(OH)D concentrations and COVID-19 severity.

25(OH)D (nmol/L)	COVID-19-Related Death			Severe COVID-19, Including Death			ICU Admission		
Time points *									
T1 (n = 37)	No	Yes	*p* value **	No	Yes	*p* value **	No	Yes	*p* value **
25(OH)D (IQR)	72 (56–86)	62 (52–70)	0.376	74 (56–86)	67 (52–72)	0.374	71 (56–85)	63 (50–80)	0.520
n	31	6		27	10		32	5	
T2 (n = 69)	No	Yes	*p* value **	No	Yes	*p* value **	No	Yes	*p* value **
25(OH)D (IQR)	61 (35–85)	53 (40–107)	0.694	60 (37–85)	68 (33–94)	0.809	60 (37–89)	59 (28–75)	0.469
n	61	8		53	16		63	6	

* Time points: T1, up to three months before the SARS-CoV-2 infection onset; T2, during the first week of hospitalisation. ** Mann–Whitney *U* test.

**Table 6 nutrients-14-01641-t006:** Vitamin D status and COVID-19 severity.

Vitamin D Status * (GRIO Guidelines)		COVID-19-Related Death			Severe COVID-19, Including Death			ICU Admission		
Time point T1		No	Yes	*p* value **	No	Yes	*p* value **	No	Yes	*p* value **
Deficiency (n)	No	30	6	1.000	26	10	1.000	31	5	1.000
	Yes	1	0		1	0		1	0	
Insufficiency (n)	No	14	1	0.368	13	2	0.153	14	1	0.629
	Yes	17	5		14	8		18	4	
Time point T2		No	Yes	*p* value **	No	Yes	*p* value **	No	Yes	*p* value **
Deficiency (n)	No	51	8	0.592	46	13	0.687	54	5	1.000
	Yes	10	0		7	3		9	1	
Insufficiency (n)	No	24	3	1.000	19	8	0.385	25	2	1.000
	Yes	37	5		34	8		38	4	

* Vitamin D deficiency was defined as 25(OH)D < 25 nmol/L (<10 µg/L); Vitamin D insufficiency was defined as 25(OH)D < 75 nmol/L (<30 µg/L). ** Fisher’s exact test.

## Data Availability

The data presented in this study are available on request from the corresponding author. The data are not publicly available due to ethical considerations, regarding personal information and respecting what was written in the generic information sheet dedicated to COVID-19 research at the Hospices Civils de Lyon (Lyon, France).

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
