# Peer review of "Vitamin D and COVID-19 Severity in Hospitalized Older Patients: Potential Benefit of Prehospital Vitamin D Supplementation"

_nutrients, 2022, doi:10.3390/nu14081641_

Round 1
Reviewer 1 Report
The manuscript entitled „Vitamin D and COVID-19 severity in hospitalized older patients: potential benefit of prehospital vitamin D supplementation” presents interesting issue, but some issues should be corrected.
General:
- Authors should in the whole manuscript clearly refer “vitamin D”, as commonly they do not specify if they mean vitamin D intake, or vitamin D status, or vitamin D supplementation – e.g. “This study aimed to assess the relationships between vitamin D (VitD) and coronavirus disease 2019 (COVID-19) severity” – we do not know what Authors mean by “vitamin D”. However, this problem is common also in the other sentences and should be corrected in the whole manuscript.
- Authors include as variables vitamin D status and declared vitamin D supplementation prior to COVID-19 incidence, but we do not know (1) what dosage of supplementation was applied prior to COVID-19 incidence, (2) what is the vitamin D intake. Those issues should be addressed within the study as its limitations.
Abstract:
Brief justification of the study should be presented
Introduction:
Lines 69-76 – authors should not describe their study in this section (such description should be presented in Materials and Methods Section). Instead, they should present brief aim of their study (similar as within Abstract)
Materials and Methods:
Authors should justify BMI cut-off of 35 kg/m2 – why such value was chosen?
It seems that Authors did not verify the normality of distribution of their data – they should do it and present the related methodology.
After verifying the normality of distribution, in case of parametric distribution mean ± SD should be presented, while for nonparametric distribution – median accompanied by minimum and maximum value.
The applied statistical test should be based on distribution
Results:
Authors should clearly present their results, as we even do not know if distribution is parametric or not
After verifying the normality of distribution, in case of parametric distribution mean ± SD should be presented, while for nonparametric distribution – median accompanied by minimum and maximum value.
The applied statistical test should be based on distribution
Discussion:
Authors should address also other potential benefits of vitamin D – e.g. for mental health of patients.
Authors Contribution:
This section should be presented
Author Response
Dear Professor,
First of all, we would like to thank you for your interest in our work and for your constructive comments.
We have highlighted the various changes in the text in yellow.
General:
- We agree with your comment. We have clarified the term "vitamin D" (status and supplementation) and made the various changes throughout the manuscript.
- We have detailed the different methods of supplementation in lines 211 to 220. A sentence has been added on the different dosages of supplementation to clarify this point. In addition, we have added in a specific paragraph concerning heterogeneities of vitamin D supplementation.
Abstract:
A sentence at the beginning of the abstract has been added for the justification of the study.
Introduction:
The presentation of the objectives of the study has been simplified and shortened in the introduction. The Flow Chart in the method section summarises also these objectives.
Materials and Methods:
- This point has been addressed in lines 142-147. “The French High Council for Public Health, in accordance with the literature and international databases, recommends that age ≥ 70 y/o and a BMI ≥ 35 kg/m² or < 18.5 kg/m² should be considered at risk of severe COVID-19.”
- The normality of the distribution of our variables was checked. We have added a paragraph on this subject (lines 138-141). The majority of the quantitative variables are not normally distributed. We therefore decided to harmonise the presentation of the results to use descriptive statistics (median and interquartile ranges). And we have carried out non-parametric tests, which are more restrictive and do not question their significance, but which may lead to a loss of power for non-significant tests.
Results:
Again, the normality of the distribution of our variables was checked. We have added a paragraph on this subject (lines 138-141). The majority of the quantitative variables are not normally distributed. We therefore decided to harmonise the presentation of the results to use descriptive statistics (median and interquartile ranges). And we have carried out non-parametric tests, which are more restrictive and do not question their significance, but which may lead to a loss of power for non-significant tests.
Discussion:
- We have added potential effects of vitamin D, other than musculoskeletal, especially regarding its potential effect on inflammation in lines 293-299.
- We have added the authors' contributions.
We hope that the changes made meet your expectations and we look forward to hearing from you in due course.
Justin BOULOY

Reviewer 2 Report
The manuscript was prepared very well. However, there are some concerns, in part important, so the review articles need revision, see below.
General comments
- Why is vitamin D supplementation necessary? What are the appropriate clinical ranges?
Materials and Methods / Results
- This is the strong part of the study; I congratulate the authors.
Discussion
- Should include some hypothesis of the possible mechanisms described from a physiological perspective or include some mechanism of action described.
- It should include some comparative discussion with other studies involving supplementation with any form of vitamin D, explaining the differences.
- Include a section on strengths
- What does this study bring back? clarify it.
Author Response
Dear Professor,
First of all, we would like to thank you for your interest in our work and for your constructive and encouraging comments.
We have highlighted the various changes in the text in yellow.
General comments:
- The indications for pre-hospital vitamin D supplementation take account the recommendations of learned societies, and with international recommendations on musculoskeletal health. These recommendations are related to the increasing prevalence of vitamin D deficiency in the general population, particularly in the elderly and for the prevention of osteoporotic fracture. Supplementation was independent of the pandemic situation. We have added a sentence in the manuscript lines 105-106 clarifying this point.
Materials and methods:
We would like to thank you for your encouraging comments.
Discussion:
- We agree with your comment. We have added elements of pathophysiology to the discussion to allow a better explanation.
- We reviewed the various data in the literature concerning the potential role of supplementation in COVID-19 and compared them to our work. Lines 330-336. Few studies have looked at pre-hospital supplementation for COVID-19, and this is one of the strengths of our work.
- We have included the strengths of our study in the paragraph titled "Limitations and Strengths of the Study". We have added a paragraph outlining what this study brings to the literature (lines 350-360).
- We have also added a sentence in the conclusion showing the contribution of this real-life study for the widespread use of vitamin D supplementation, particularly in this under-supplemented population.
We hope that the changes made meet your expectations and we look forward to hearing from you in due course.
Justin BOULOY

This manuscript is a resubmission of an earlier submission. The following is a list of the peer review reports and author responses from that submission.
Round 1
Reviewer 1 Report
Dear authors,
I put all my comments in the article attachment. What I consider more serious is the non-indication of the submission
of the project under analysis to an ethics committee. I consider it necessary to consider the standardization and proper use of key terms in this article, such as the definition of what a geriatric population is. The BMI cut-off points should also be reviewed and standardized. But in general, it is an important and interesting work.

Reviewer 2 Report
I read with interest this paper on vitamin D and Covid-19. Although the results seem positive I think there are a few things that have not been addressed and need addressing in the paper
- Ethical approval: It is not clear if ethical approval was obtained for the study. It is not enough to "tell" patients about the study. Was consent obtained?
- It is not clear if patients had vitamin D as a one off dose or regular medication. Compliance of treatment and dose has not been mentioned. We know that vitamin D taken as a one off has a half life of 30-45 days. So if treatment was taken 3 months prior there would not be any vitamin D in the body.
- The tables are confusing. N=228 but in the table n=417 (Table 1). This is because the authors have combined outcomes. The term 'severe covid' is not a standard term. How was this defined. What was the criteria for ICU admission. Was there a age related cut-off for ICU admission
- therefore it would be better to look at difference in outcomes according to age ranges
- Serum level of Vit D: If the authors are quoting both ug/l and nmol/l then these should be stated for all the ranges in the methods.
- This is a snap shot of a larger study. How were the patients selected for this study. was there a selection bias?
- The authors do not mention about the calibration of the assays and quality control checks (eg DEQAS)
- The authors mention T1 and T2 time points for Vit D measurement. How is this impacted on the study?
- Ultimately there are many studies on vitamin D. the authors need to justify how this is different from those published so far and what does this add to that what is already known.